# Combining Multichannel RSSI and Vision with Artificial Neural Networks to Improve BLE Trilateration

**DOI:** 10.3390/s22124320

**Published:** 2022-06-07

**Authors:** Sharareh Naghdi, Kyle O’Keefe

**Affiliations:** Position, Location, and Navigation (PLAN) Group, Department of Geomatics Engineering, Schulich School of Engineering, University of Calgary, 2500 University Drive, N.W., Calgary, AB T2N 1N4, Canada; sharareh.naghdi@ucalgary.ca

**Keywords:** advertising channels, BLE, trilateration, artificial intelligence, localization, human body shadowing

## Abstract

The demands for accurate positioning and navigation applications in complex indoor environments such as emergency call positioning, fire-fighting services, and rescue operations are increasing continuously. Indoor positioning approaches apply different types of sensors to increase the accuracy of the user’s position. Among these technologies, Bluetooth Low Energy (BLE) appeared as a popular alternative due to its low cost and energy efficiency. However, BLE faces challenges related to Received Signal Strength Indicator (RSSI) fluctuations caused by human body shadowing. This work presents a method to compensate RSSI values by applying Artificial Neural Network (ANN) algorithms to RSSI measurements from three BLE advertising channels and a wearable camera as an additional source of information for the presence or absence of human obstacles. The resulting improved RSSI values are then converted into ranges using path loss models, and trilateration is applied to obtain indoor localization. The proposed artificial system provides significantly better localization solutions than fingerprinting or trilateration using uncorrected RSSI values.

## 1. Introduction

The position estimation of a target in outdoor environments is widely solved by employing Global Navigation Satellite Systems (GNSSs). The received GNSS signals are weak or not reliable inside the buildings. In the past few years, GNSS has developed to serve in indoor areas. In [1], the authors present an overview of GNSS-based indoor location technologies. The paper discusses the attenuation of the GNSS signals under various conditions such as a wooden structure building, single-family residence, and large sports facility. GNSS signals are attenuated indoors by 10–30 dB compared to the outdoors depending on the building material. Considering the fact that people spend most of their time in indoor environments where GNSS signals are not usable in a satisfactory manner, the demand for reliable indoor localization applications has rapidly increased. Numerous Indoor Positioning Systems (IPSs) based on a wide range of technologies have been studied [2]. Among the technologies, Radio Frequency-based (RF) indoor positioning is prevalent, freely available in the indoor areas, and in most cases, supported by current mobile devices. However, RF technologies have limitations, including vulnerability to signal interference, multipath and signal attenuation, requirements for clock synchronization between transmitters and receivers, or requirements for calibration.

RF positioning systems rely on three different specific physical characteristics of radio signals: (i) the power of the propagated signal, (ii) the propagation time, and (iii) the direction of the propagation wave and are often classified into three corresponding categories: (i) Received Signal Strength Indicator (RSSI), (ii) Time of Flight (TOF), and (iii) Angle of Arrival (AOA) methods [3].

Approaches that employ RSSI measurements are considered the simplest, since they do not necessitate any hardware modifications, system timing coordination, or synchronization between transmitters and receivers. Although the RSSI method is one of the most widespread approaches, there are still some challenges that need to be addressed. In general, RSSI position accuracy is poor because the signal strength can be affected by multipath, interference, and shadowing caused by various factors such as the presence of obstacles, the number of reflective surfaces, and the overall dynamics of the environment.

RSSI-based ranging and fingerprinting are two common methods that rely on the RSSI measurements for positioning. Positioning that utilizes fingerprinting has the potential to achieve high accuracy, provided sufficiently dense training data are available; however, this process is time consuming and does not adapt well to environmental changes. RSSI-based ranging needs a path loss model to estimate ranges from the RSSI values and then applies trilateration to compute a position. A general overview of the wireless ranging positioning methods is available in [4].

Wi-Fi technology is the most commonly employed in RF-based indoor positioning; however, in recent years, the smart device market has increasingly relied on Bluetooth for short-range device-to-device communication, which makes this technology a viable alternative for indoor positioning [5,6]. The availability of existing Wi-Fi infrastructures in indoor environments makes it one of the best signals of opportunity for indoor localization. However, the placement of access points is not necessarily optimal for localization. In [7], the BLE-based localization was compared with the Wi-Fi localization. Their results show that BLE is more accurate than Wi-Fi by about 27% with the same number and location of access points. Another comparison between BLE and Wi-Fi fingerprinting is presented by [8]. They show that BLE fingerprinting has a tracking accuracy of less than 2.6 m 95% of the time by using a dense distribution (1 beacon per 30 m^2^) and less than 4.8 m using a lower density distribution (1 beacon per 100 m^2^). Meanwhile, Wi-Fi can achieve about 8.5 m 95% of the time in the same environment. They conclude that increasing the number of beacons decreases the positioning error, but after 8–10 beacons, there was no further improvement.

Bluetooth has received more attention since 2010 when the Bluetooth Special Interest Group (SIG) introduced the Bluetooth Low Energy (BLE) standard with the Bluetooth core specification version 4.0. BLE is a very low power, low cost, low complexity, and low maintenance technology [9]. Depending on the connection interval, battery-powered BLE modules are able to last for 1–2 years [10].

While Bluetooth was originally developed to replace cables connecting personal electronics, BLE beacons were meant to broadcast highly localized information to passing Bluetooth devices. The transmission power of BLE beacons is adjustable, and they reliably transmit data up to 30 m [11]. BLE operates at a frequency band of 2.4 GHz, which is divided into 40 channels with 2 MHz spacing [12]. In addition, BLE has been used widely in different applications such as building emergency management and occupancy estimation [13], occupancy movement tracking patterns in office spaces [14], model-based localization and movement tracking [15], and smart grid applications and home energy management [16].

Out of these 40 channels, three advertising channels (labeled 37, 38, and 39) are reserved to continuously broadcast advertising messages. Initially, the proposed application for the BLE beacons was proximity marketing, which advertises marketing messages to mobile devices close to a particular position in shopping centers, museums, hotels, stadiums, exhibition halls, etc. Examples of these marketing messages include relevant information, related news, and special offers.

RSSI values can be obtained from these three channels by users in the range of the BLE beacons. Since each channel has a different channel gain and multipath, the RSSI values from each of the advertising channels differ [8]. Most existing research works have considered the RSSI values from all three channels together to achieve the aggregate signal [5,6,17,18,19], which contains more fluctuations when compared to each of the individual channels. To achieve more accurate results, others have considered RSSI values from separate channels, although these studies have used complicated algorithms [20] or extra hardware [21,22]. In [23], the BLE version 5.x is used, which is capable of advertising on three primary channels and a single auxiliary packet on one of the data channels. This extended advertisement allows measuring RSSI on all 40 channels. Their results showed that using the Geo-N algorithm, the mean localization error difference between 40 channels and three channels is improved by 0.47 m and 0.35 m, with and without channel information, respectively. This improved the accuracy at the cost of a larger number of measurements collected across 40 different channels and consequently a longer required time for the localization to be calculated. The application of BLE with three advertising channels provides the advantage of redundancy between the RSSI values in addition to time-efficient measurements and data collection.

Signal fading may either be due to the interference from multipath propagation or shadowing from obstacles, and it is often neglected in the available models. However, to properly model radio propagation for RSSI-based ranging, all the obstacles between the transmitter and receiver should be considered. A common signal attenuation source is the human body, which can shadow or fully obscure the signal path. A human body can be detected by many different approaches and technologies such as vision, distortion, irregularities in the signal, etc. The three BLE advertising channels follow the same pattern during a human body blockage. When a human body blocks the signal, all three channels drop, and when a human body leaves the area, all three channels return to their former values simultaneously [24,25] as opposed to multipath fading that affects the channels individually.

Human body detection plays a critical role in calculating the human body shadowing effect. An efficient combination of camera measurements and the RSSI model is used in [26], which trains RSSI range models to adapt to the conditions of a particular environment for target tracking. However, here, the information from vision is used to detect people for compensation shadowing effects and not for tracking purposes.

In computer vision research, objects such as human bodies can be detected in two main categories: (i) hand-crafted features [27] and (ii) learning-based methods [28]. The first category relies more on pre-designing descriptors including Haar [29], Local Binary Pattern (LBP) [30], Histogram of Orientated Gradients (HOG) [31], Scale-Invariant Feature Transform (SIFT) [32], etc. The drawback of the hand-crafted methods is the requirement to extract features manually from the raw data by using specialized algorithms. In contrast, learning-based methods can automatically learn from the raw data often with less computational time and more reliable performance. Deep learning-based methods are one of the most powerful classes of object detection algorithms [33]. A Convolutional Neural Network (CNN) is a type of deep learning technique that has been widely used in human body detection in challenging indoor environments [34]. Among all the deep learning approaches, the highest accuracy of human detection belongs to the RetinaNet method [35].

To obtain occupancy information, there are other techniques based on using a BLE beacon network. The authors in [36] applied an SVM algorithm for occupancy estimation using BLE beacons for emergency management. Using a network of BLE beacons to record the RSSI values of neighboring devices to infer the occupant’s zone-level location was proposed in [37]. Another regression model and decision trees (random forest) based on BLE beacon networks were adopted in [38]. All these approaches require the occupants to carry permanently connected devices which have limitations in privacy concerns, with users forgetting to turn on the Bluetooth of their devices and the fact that they had to carry the sensor with them all the time. In our approach, however, the only person that needs to wear the device is the user, and other occupants do not need to be involved at all.

RSSI values can be mathematically related to the range by empirical or Artificial Intelligence (AI) algorithms [39]. AI models have been considered a powerful tool to learn from the observed data in real environments and model the complex relationships between the input and output values. Empirical models perform well in terms of processing time and memory efficiency, although they are less compatible with sudden changes in the propagation environment [40,41]. Flexible neural network solutions can be used to model the relationship between the predicted and measured RSSI values and have been demonstrated in [42,43,44]. Their advantage over threshold-based detection methods is in their ability to adapt based on the observed data rather than on analytical and theoretical models of a system [45,46]. Moreover, previous studies [20,21,22] that used BLE RSSI values from separate channels did not investigate using AI approaches to correct the RSSI for human body blockages. In our previous work [24], we demonstrated that using the three advertising channels with Artificial Neural Networks (ANNs) to detect human obstructions provided more accurate results than when using all the available RSSI observations in the aggregate. However, other available sensors on mobile devices provide the opportunity to collect more information about the propagation environment. This information can prepare more accurate input parameters with proper weight for ANNs.

The potential applications of this work can be the support of the firefighters’ movements or the tracking of the emergency staff, using the BLE signal with an AI-based algorithm, which is augmented by the vision information.

This study is the first demonstration of the combination of BLE RSSI, wearable camera, and ANN to detect and correct RSSI values for human body obstructions. The potential applications of this work can be the support of the firefighters’ movements or the tracking of the emergency staff, using the BLE signal with an AI-based algorithm, which is augmented by the vision information. The objective of this paper is to investigate the application of simple BLE trilateration positioning in real indoor environments in which human bodies are present and can affect the signals using an ANN to correct the observed RSSI measurements. The proposed neural network algorithm is implemented to detect and correct for the presence of human obstacles using observations from the three BLE advertising channels and vision information captured with a wearable camera as an additional source of information. The results of this method are then compared to our previous RSSI-only ANN results [24], fingerprinting, and trilateration using uncorrected RSSI values.

The remainder of this paper is organized as follows. Section 2 provides the relevant background of each of the technologies. Section 3 describes their integration. Data collection is described in Section 4, and the results are discussed in Section 5.

## 2. Methods Employed

This section reviews the methods used in this paper that are combined in Section 3, specifically BLE, ANN, path-loss models, and vision techniques.

### 2.1. Bluetooth Low Energy

BLE is a recent wireless communication technology that is emerging as a standard for indoor positioning based on a 40-channel frequency hopping scheme. For discovery services, BLE uses three advertising channels: 37 (2402 MHz), 38 (2426 MHz), and 39 (2480 MHz). Figure 1 depicts how the BLE channels are positioned in the frequency band. The first channel, 37, is centered at the frequency of 2402 MHz, while the last one, the 39, is centered at 2480 MHz. Channels from 0 to 36 are assigned for data transmission. It should be noted that the three advertising channel numbers are not sequential and include the lowest and highest center frequencies.

BLE can operate in two modes. The first communication mode is the traditional connection-based mode, which needs to pair the transceivers with the connection interval. The second communication mode is the connection-less mode, in which the transmitter is broadcasting to a receiver and the transmitters are unaware of the number of the advertising packets received by the receiver. One of the most important features of the broadcaster is the advertising interval or the rate at which the advertising packets are sent. On the other hand, scan intervals and scan windows represent the rate that the scanner turns on and the time it keeps on scanning per each scanning interval. The scan interval and scan window sizes have a deep impact on power consumption. Most importantly, the transmitter sends advertising packets on all three channels sequentially at a relatively high rate while the receiver scans one channel at a time at a lower rate.

The duration of the advertising interval and scanning windows can cause multiple measurements of one channel in some periods of scanning. Remaining asleep during broadcast intervals helps the BLE system to achieve an optimal power consumption; however, a shorter broadcast interval increases the number of broadcasted packets and the accuracy of their readings at the expense of additional power consumption.

As each channel has a slightly different carrier frequency, each BLE advertising channel will have distinct propagation characteristics owing to varying channel gain and multipath fading. In Figure 2, when the channels are considered separately (the first 50 samples), small fluctuations are visible in each, while the fluctuations appear larger when considering them in the aggregate mode (the next 50 samples).

### 2.2. Distance Model and Trilateration

The power density of the signal attenuates as it propagates through space as well as objects. The most commonly used distance model is a standard log-distance path loss model:(1)$$\mathrm{RSSI}=RSSI\left(d_{0}\right)-10n{\;\log}_{10}(\frac{d}{d_{0}})+X_{\sigma }$$
where $RSSI\left(d_{0}\right)$ represents the RSSI value at the reference distance $d_{0}$, $X_{\sigma }$ and n represent the observation error and path loss exponent value, respectively, and d is the distance between the transmitter and the receiver. In free space, n is 2, while it is often greater because of the other sources of attenuation and can be less than 2 in waveguides. Usually, parameter $d_{0}$ is fixed to 1 m, and $RSSI\left(d_{0}\right)$ becomes the average measured RSSI when the receiver is 1 m away from the transmitter. The path loss exponent n, which is related to the wireless environment along with $RSSI\left(d_{0}\right)$, can be determined either by fitting a line to training measurements or by choosing a standard value. Theoretically, n should be constant; however, in reality, the BLE transmit power has time-varying characteristics, and the path loss exponent is dependent on the environment.

Trilateration algorithms use the distance $d_{m}$, estimated from the received RSSI values from all transmitter nodes to compute the position of the single intersection. For overdetermined trilateration with errors, non-linear parametric least squares are the standard method used to find a solution that minimizes the mean squared error of the residuals.

If m transmitters with known coordinates ($x_{Tx_{1}},y_{Tx_{1}}),\;\left(x_{Tx_{2}},y_{Tx_{2}}\right),\ldots ,\;\left(x_{Tx_{m}},y_{Tx_{m}}\right)$ are deployed, and the receiver has an unknown location$\;\left(x_{Rx},y_{Rx}\right)$, the m distances are related to the unknown positions as:(2)$$d_{m}=\sqrt{{\left(x_{Tx_{m}}-x_{Rx}\right)}^{2}+{\left(y_{Tx_{m}}-y_{Rx}\right)}^{2}}$$

It should be noted that knowing the position of the transmitters is not required in the fingerprinting technique, which is one of the advantages of this method.

In this work, we are considering 2D positioning only and have constrained the height of the receiver and have corrected observed distances for their vertical components. The state vector x is given by:(3)$$𝑥={\left[x_{Rx}\;y_{Rx}\right]}^{T}$$
where $x_{Rx}$ and $y_{Rx}$ are the 2D position components in the horizontal plane (East and North). The observation model is:(4)$$z=h\left(x\right)+v$$
where $z={\left[d_{1},\;\ldots ,d_{m}\right]}^{T}$ are the distance estimates from the propagation model, v is the vector of measurement errors which is modeled as a Gaussian distribution, with a covariance matrix $R=E\left(v,v^{T}\right)$, and $h\left(x\right)$ is a vector where each element is an instance of Equations (3) and (5). To linearize the non-linear measurement model, a Taylor series is applied:(5)$$H_{m}={\left.\left[\begin{array}{cc} \frac{-\left(x_{Tx_{m}}-x_{Rx}\right)}{d_{m}} & \frac{-\left(y_{Tx_{m}}-y_{Rx}\right)}{d_{m}} \end{array}\;\right]\right|}_{x=x_{0}}$$
where $x_{0}\;$is the point of expansion. The result is the design matrix H that contains information regarding the geometry of the measurements. The misclosure vector (δz) is the difference between the true measurements $\left(z\right)$ and the measurements estimated from the current states ($x_{0}$):(6)δz=H.δx+v 

The least-squares solution for the error in $x_{0}$, which is applied to the original state vector to correct it to the next solution $x_{1}$, is given by:(7)$$\delta x={\left(H^{T}R^{-1}H\right)}^{-1}H^{T}R^{-1}\delta z$$

To make this distinction more explicit, the initial state estimations are then updated as follows:(8)$$x_{1}=x_{0}+\delta x$$
since the model is non-linear, iteration is used to converge to a final solution $\hat{x}$, which yields no further improvement with additional iteration.

### 2.3. Neural Network Algorithms

ANNs consist of several simple and highly interconnected processing neurons set up in layers. Multilayer Perceptron (MLP) and Radial Basis Function (RBF) neural networks are two of the basic and well-known types of neural networks with a wide range of applications in many areas of estimation and decision making, including indoor positioning. The multilayer perceptron model uses a linear weighted function for each neuron. Multilayer perceptron models have at least one hidden layer and can handle non-linear terms. Since the relationship between RSSI and distance is non-linear and a single layer cannot model non-linear terms accurately, an MLP is considered.

RBF neural networks use a radial basis activation function. The activation of a hidden unit is identified by the distance between the input vector and a prototype vector. Hidden neurons are dynamically generated during the training procedure to achieve the desired performance. The number of basis functions is equal to or less than the number of input data sets. In this research, a supervised learning method with an error backpropagation algorithm is employed. In the backpropagation algorithm, at first, the input vector is propagated with constant weights and biases through a forward pass, and the output is produced. Then, synaptic weights and biases are adjusted by using the error signal that propagates backward to minimize the cost function of the neurons in the output layer.

MLP networks consist of a single input layer, at least one hidden layer, and a single output layer. The output of each neuron is described by the following:(9)$$y=\varphi \left(\overset{n}{\underset{k=0}{\sum }}w_{k\;}x_{k\;}\right)$$
where n∈N is the number of neuron inputs, $x_{k\;},w_{k\;}\in R$ are the input value and its weight, respectively; at the *k*th neuron, y is the neuron output and $\varphi \left(x\right)$ is an activation function. Figure 3 illustrates the structure of an MLP neuron network and a single neuron model inside an MLP. Each activation function receives the sum of the weighted inputs plus a bias term (Ɵ).

The activation function is a mathematical gate in between the inputs and outputs of a neuron which can be a step function (i.e., output is active if the input value is greater than a threshold value), a linear function (i.e., the output is the input times some constant factor) or a non-linear function.

The non-linear functions allow the model to map the complex relationships between the inputs and outputs, which are essential for the learning and modeling of complicated real data. The most common non-linear activation function is Logistic (also known as the Logistic Sigmoid):(10)$$\varphi_{log\;}\left(U_{k\;}\right)=\frac{1}{1+\exp\left(-U_{k\;}\right)}$$

RBF is an ANN technique that identifies the activation of a hidden unit by the distance between the input vector and a prototype vector during the training (Figure 4). Each neuron in the hidden layer consists of a radial basis function, and the output layer is a weighted sum of the outputs from the hidden layer. The hidden and output layers apply a non-linear and a linear transformation, respectively. The training procedures in the RBF networks can be significantly faster than the training procedures in the MLP networks [47].

There are two stages in the training procedure of an RBF network. The first stage involves the determination of the mean value and distance from the center of the activation function using the input data by unsupervised training methods. In the second stage, the output layer weight vector is determined. In RBF, the hidden layer uses a set of Gaussian functions, known as radial basis functions, which is given by:(11)$$\varphi \left(x,\mu \right)=exp\left(-\frac{{\left(x-\mu \right)}^{2}}{2d^{2}}\right)$$
where μ is the center of the Gaussian function (i.e., the mean value of (x)), and 𝑑 is the distance from the center of the Gaussian function. The output of each hidden unit is based on the distance of the input from the center of the Gaussian radial function$\;\varphi \left(x,\mu \right)$. Subsequently, data points closer to the center of the radial basis function have more effect on the results. This effect can be adjusted by controlling the distance (d). Parameters (d) and (μ) are defined and adjusted separately at each RBF unit during the training procedure. Layer 3 or the output layer is a weighted linear combination of the outputs from the hidden layer:(12)$$\mathrm{output}=\underset{i}{\sum }(\varphi_{i}W_{i})$$

### 2.4. Human Body Detection

The human body is one of the non-negligible sources of propagation loss. The human body shadowing effect can be caused by the user of the device as well as other people close to the device. In this paper, we propose to use both the RF signals and the vision information to determine the number of people blocking a signal. Specifically, the method introduced in our previous work [24] using ANN to detect and correct human body shadowing using three channels of RSSI measurements is augmented with additional input information obtained from a wearable camera.

Visual human body detection is a computer vision problem that deals with the detection of a human body in a digital image. Deep learning algorithms have become popular due to their powerful ability in detection tasks. Deep learning frameworks often use one-stage or two-stage detectors. In two-stage detectors [34], a proposal generator generates potential objects as a set of rectangle bounding boxes to extract features from each proposal. Region classifiers predict the category of the proposed region. However, one-stage detectors [35] predict directly each location of the feature maps. In order to estimate the number of people in an image, a robust one-stage object detector, RetinaNet, was employed and adapted [35].

In this method, a combination of Feature Pyramid Network (FPN) and ResNet was used as the backbone architecture (Figure 5). Two subnets of classification and box regression were used. These two subnets are used for the classification and bounding box regression to perform convolution, respectively. The backbone’s responsibility is to compute a convolutional feature map over an entire input image.

## 3. System Design

The general overview of the proposed algorithm is demonstrated in Figure 6. A set of m transmitters is deployed in fixed positions. The person carrying the mobile receiver to be positioned is also wearing a camera. The camera images are processed by an implementation of the RetnaNet algorithm to determine the count of detected humans in the image. To create a memory of the previously obtained values of the RSSI measurements, a common method is to use a sliding window of past sequence values as inputs to the ANN. Static memory is then provided for the network to map inputs to outputs, depending on the prior information. The sliding window technique gives the opportunity of learning sequential patterns of the past N values of the RSSI from the three BLE advertising channels. The method is, in fact, looking for abrupt changes in the RSSI due to obstructions that are distinct from gradual changes that are a result of changes in range. This neural network method is trained to notice the sudden simultaneous fluctuation of all three channels as an obstacle. However, a sudden fluctuation of only two advertising channels would not be considered an obstacle. Moreover, a deep learning algorithm is used for estimating the population density of the captured images by an expert user’s camera.

Sliding windows of the past N values of this count and the past N values of the RSSI values from the three BLE advertising channels are then given as input to the ANNs, which are each responsible for computing corrected RSSI values that can then be used to generate three ranges (one based on each BLE channel) to that transmitter. These ranges are then used to trilateration the user position.

The ANNs are trained by collecting a large set of RSSI values and imagery at a number of known locations with and without additional people present to block some of the signals. More details of the system, data collection, and databased are presented in Figure 7. The deep learning algorithm RetnaNet, used in the image processing block, is not a part of the main ANN system and is trained separately.

Here, we are assuming the camera is worn by an expert user, for example, a first responder, as opposed to a normal user who would likely not have a wearable camera and would have to rely on RSSI measurements only. The problem of visual detection of occlusion through the user from behind has not been addressed in this study; however, more redundancy on the RSSI values in AI inputs and considering all APs together make the system able to detect the blockages and correct that situation. The number of people identified in the imagery is then passed to the ANNs along with the BLE RSSI values. The ANNs are then tasked to output corrected BLE RSSI as would be observed if no additional people were present to block the signal.

The step-by-step algorithm for correcting the RSSI measurements is explained briefly as follows:Obtain the RSSI values for each channel corresponding to each of the transmitters when no people are blocking any signals.Repeat Step 1 with 1 or 2 people blocking some of the signals.For training, the inputs are selected from Steps 1 and 2 randomly, while the outputs can be only selected from Step 1, since they represent the RSSI values with no blockages. Most (70%) of the measurements in Steps 1 and 2 are used to train the ANN system.Evaluate and test with 30% of all measurements from Steps 1 and 2.

For separating training and testing data sets, there is no fixed rule [27]; however, the data could be split between training, testing, and validation in the ratio of 70%, 15%, and 15%, respectively. As with all AIs, the prediction of the neural networks highly depends on how well they learn the concepts from the training data and apply them to the testing samples. Reduction in the generalization ability can occur from overtraining, while the expansion in generalization ability can cause undertraining. To avoid this issue, a partition of the data is required to be specified for the validation data set as it plays a vital role alongside the training and testing data sets.

Figure 8 illustrates the schematical difference between the sampling rate in the RSSI values in three channels and the captured images. More details of the camera specifications are discussed in the next section. The sliding windows update at the rate of the RSSI measurements and the count obtained from the vision block is fed to the ANNs at the same rate even though the frame rate of the camera is lower. This means that the current count is repeated in the sliding window until a new image is required.

In [24], the Mean Square Error (MSE) of training (system output vs. desired response) was compared as a function of window size to choose an optimal sliding window. Although the processing time increases, the training error decreases as the window size is increased. The training errors were stable around a window size of 10. For comparison purposes, the same value is adopted here.

## 4. Experimental Setup

Two different BLE development kits were selected. (i) The DWM1001-DEV (Decawave Ltd., Dublin, Ireland) modules that include an nRF-52832 (Nordic Semiconductor, Trondheim, Norway) BLE radio, a LIS2DH12 (STMicroelectronics NV, Amsterdam, Netherlands) accelerometer, and a DW1000 UWB chip (Decawave Ltd., Dublin, Ireland) was chosen as the transmitter because of its low power consumption, small size, very low cost, battery-operated power, and easy deployment on walls. Each transmitter was configured to send the BLE advertising information at an interval of 20 ms.

(ii) The nRF52840 development kit (Nordic Semiconductor, Trondheim, Norway) which includes the nRF-52840 (Nordic Semiconductor, Trondheim, Norway) BLE radio, four buttons and four LEDs for the user interaction, a flash memory, PCA10056 chip, and a Near Field Communication (NFC) antenna was selected as a receiver. It is included in the PCA10056 development board that provides onboard debugging as well as the programming solution. The nRF52840 development kit was selected as the receiver because it could provide full chip-level access to BLE and debug interfaces to develop and configure a data-logging application. The receiver was configured to measure the RSSI values on all advertising channels with a scanning interval of 50 ms.

To test the system in a complicated and large environment, experiments were carried out in a larger electronics lab on the 3rd floor of a multi-story university building. The dimensions of this room were approximately 8 m by 16 m (Figure 9). This lab contained numerous workbenches, shelves, and storage cabinets.

The 2D plan of the lab area is illustrated in Figure 10, in which 34 reference points with known locations were established. Four DWM1001-DEV modules with known locations served as the BLE transmitters (orange squares). A GOPRO HERO 7 high-resolution (4000 × 3000 pixel) digital camera (GoPro, Inc., San Mateo, CA, USA) and the receiver were carried by the test subject in the lab area. Both (camera and receiver) were kept at the same height. The camera was set to record images at a 1.0-s sampling rate.

The goal was to evaluate the system in the complicated lab and compare it with the proposed system in [24], fingerprinting and trilateration using uncorrected RSSI, as well as assess the proposed system in non-trained locations in the lab. This was completed by conducting two test scenarios. During the first scenario, the neural network was trained and tested using 34 reference points, while in the second scenario, the trained network from the first scenario was used to test the proposed system in additional locations in the same lab that were not occupied during the training phase.

In the first scenario, the receiver and camera were moved through the 34 reference points (blue circles in Figure 10) to collect 215 RSSI samples at each reference point in the absence of a human body as an obstacle, 70 RSSI samples were collected with one human body obstacle and 70 more RSSI samples with two human bodies obstructing at least one signal at each reference point. Most (70%) of the collected data was randomly selected for training the system, and 30% was reserved for testing. The training output was the RSSI values on each reference point with no person in the room, even the user. Based on the dynamics of the test environment, the image sampling rate was one image per second, and 16 images were captured at each point. Note that the objective is to detect people in low rate imagery rather than to track them in high rate video, which is the subject of a large body of research [48,49]. In this work, we assume that both observation types were continuously available at their respective observation rates, and we did not account for the case where the image data cease to be available.

The second scenario was intended to evaluate the already trained network from the first scenario in some unknown and untrained locations referred to as blind test points. Five blind test points were randomly selected in the same lab (yellow circles in Figure 10), and 200 RSSI measurements were collected at each with random appearances of people during each occupation. All gathered data in the second scenario were used for testing without a training phase for these blind points. A total of 150 images were collected for the second scenario (30 images at each point).

## 5. System Verification

The test scenarios included both line-of-sight (LOS) and non-line-of-sight (NLOS) blockages, but only LOS blockages were included in the training. Two additional tests were conducted to assess how NLOS blockages could affect the RSSI values as a function angle with respect to the LOS. In the first test, a human obstacle started in the LOS and then moved to four additional positions at angles between 45° and 180° off the LOS. In this test, the transmitter and receiver were separated by 2 m, and the blocking human was located 1 m from the transmitter; the first angle in the LOS and the remaining four angles were at NLOS positions, as shown in Figure 11. The maximum effect to the body shadowing was observed at 0°, but an effect could also be observed at 45°, while 90°, 135°, and 180° showed very little effect. The mean and standard deviations of all histograms in Figure 11 are summarized in Table 1.

A second RSSI experiment was conducted to verify the influence of the shadowing effect of the user and a second person when the user (and receiver) was not facing the transmitter (NLOS). Results are presented in terms of the RSSI values on channels and aggregate mode. Figure 12 shows two scenarios in which the transmitter and receiver are 2 m apart. In the first scenario (Figure 12a), the user (and receiver) was facing sideways with respect to the transmitter. A second human body was then present for 30 s in front of the receiver (but not in the line of sight). This second human left for one minute and then returned to the same spot for 30 s. The second scenario (Figure 12b) represents the first scenario with the user and receiver facing away from the transmitter. The information in Figure 12 is summarized in Table 2. The results demonstrate a significant body shadowing effect on all channels of the RSSI measurements even when the receiver does not face the transmitter. However, this effect is less in the 180° rotation scenario.

Both ANNs include three layers: input, hidden, and output layers. The neurons in the input and output layers are dependent on the configuration of the proposed system design. In both ANN methods (MLP and RBF), the number of input nodes depends on the sliding window length, 10 samples of vision, and 10 RSSI samples on each channel, requiring a total of input 40 nodes.

RBF has one hidden layer, but MLP can have a variable number of the hidden layers. MLP was investigated for one and two hidden layers with 1 to 100 neurons per layer. The optimal number of the hidden neurons for the MPL could be determined by observing Figure 13 in which the standard deviation of the output is plotted as a function of both the number of layers and the number of neurons per layer. Two hidden layers provided more accurate prediction solutions than one. There is also no improvement in the accuracy beyond 30 neurons per layer. As a result, an architecture with two hidden layers and 30 neurons per hidden layer was adopted for MPL. The same number of neurons was adopted for the single hidden layer using RBF.

In order to avoid overtraining, one can evaluate loss function per iteration for training, test, and validation datasets. In this case, the loss function is the MSE as a function of the number of training iterations. This shows that the network has arrived at the best learning and the lowest error after a certain number of iterations. To validate the performance of the proposed system, 12,070 samples of RSSI were collected from each transmitter on each of the three channels (7310 line of sight, 4760 obstructed).

From 12,070 measurements, 8449 were employed to train the network, 1810 were reserved for validation purposes, and the remaining 1811 non-training observations were used to test the system performance. Figure 14 shows the training, validation, and testing performance for observations from transmitter #1 in terms of the mean squared error (in RSSI) as a function of the number of iterations. The model could converge within 39 iterations, and model weights were chosen based on this epoch.

## 6. Experimental Results

With the ANNs properly configured, the ability of the proposed system to detect humans is assessed in this section. Moreover, the range and positioning errors in both scenarios are investigated.

### 6.1. Detecting Humans and Correcting RSSI

The ability of the vision system to correctly detect a human body in the image is evaluated in terms of the number of missed detections and false alarms. Table 3 shows the number of correctly or incorrectly detected people in the images using the deep learning RetinaNet system.

As described earlier, 16 images were captured in each reference point in test case #1 for a total of 544 images (204 obstructed by one or two persons and 340 unobstructed).

The RetinaNet system was able to correctly classify 92% of the images where a human body was present, whereas 8% were missed detections. When the human body was not present, only 10% of these images were false alarms, and the remaining 90% were correctly classified as unblocked situations.

To illustrate the ability of the proposed system to correct RSSI values, data from transmitter 1 observed at reference point 26 are illustrated in Figure 15. The raw training RSSI values are shown in the upper subplot, and raw and MLP corrected testing data are shown in the lower subplot. From the initial samples, with no obstructions, the advantage of considering the three advertising channels separately is obvious. With one and two obstructing people present, the uncorrected RSSI values are lower and more variable in all three channels. The output of the ANN (the corrected RSSI) clearly demonstrates its ability to detect and correct the effect of the human body.

### 6.2. Range and Position Estimation: Scenario #1 (at Training Locations)

The system is then evaluated in terms of range and positioning in the first scenario. Figure 16 shows the distance errors, before and after correction, from transmitter 1 to each of the 34 reference points.

Figure 17 summarizes these results for the distances from all four transmitters using the standard deviation of the distance error. The proposed method (MLP and RBF plus vision) is compared to not using vision for training or testing (the method from [24]), and both are compared to using the uncorrected input RSSI by channel and in aggregate. Similar to [24], empirical path loss models were used for their improved performance in indoor environments as opposed to selecting a standard value for the path loss exponent. As expected, smaller fluctuations in range values were observed after correction with MLP and RBF augmented by vision offering the best performance with MLP slightly outperforming RBF.

The positioning errors in the east–west and north–south directions for the first test scenario (Figure 10) are plotted in Figure 18. In addition to the MLP and RBF with and without vision information [24], fingerprinting and trilateration using the uncorrected RSSI measurements (here called classic) are shown for comparison.

The positioning errors using uncorrected RSSI (blue line), are better than 10 m in east–west, and 7.2 m in north–south for 90% of the estimates.

The fingerprinting algorithm (black line) produced results with 90% of the points better than 6.8 m in east–west and 5.5 m in north–south.

The positioning errors using RSSI corrected by the ANN system [24] without the vision information (labeled AI) are shown for MLP (red dashed line) and RBF (green dashed line). Most (90%) of the points are better than 4.7 m and 5 m in east–west, offering a reduction of 31% and 26% over fingerprinting, and 53% and 50% over classic trilateration, respectively. Less improvement in the north–south position is observed with 90% of the points better than 4.9 m and 5.4 m, a reduction of 11% and 2% over the fingerprinting, and 32% and 25% over classic trilateration, respectively.

Adding the vision information results in a significant improvement. MLP (red solid line) and RBF (green solid line, both labeled AI + Vision) provide east–west positions better than 2.9 m and 3.5 m 90% of the time. These are 57% and 48% better than fingerprinting and 71% and 65% better than classic trilateration, respectively. The corresponding north–south values, which are better than 2.4 m and 4.1 m 90% of the time, are an improvement of 56% and 25% over fingerprinting and 67% and 43% over classic trilateration, respectively. All four ANN methods perform well, but MLP outperforms RBF both with and without the additional vision information. As a result, only MLP is tested in the second scenario.

### 6.3. Positioning Estimation: Scenario #2 (at Blind Test Points)

To evaluate the proposed system at untrained locations, five points in the lab environment that were not occupied during the training were selected randomly for testing. Figure 19, Figure 20 and Figure 21 show the ability of the proposed algorithm at one of these locations.

Raw and corrected RSSI values are shown in Figure 19, while the position solutions for each of the different methods are presented in Figure 20. The training reference points are shown in Figure 21 as pink squares, while the blind point is the red square. The uncorrected RSSI values provide very poor results as shown by black circles, while fingerprinting offers positions near training data (red circles). In contrast, the MPL ANN method was able to provide a reasonably precise position (blue circles).

Figure 21 summarizes the positioning results for all five blind test points comparing maximum and Root Mean Square Error (RMSE) using MLP with RSSI and vision, fingerprinting, and trilateration from uncorrected RSSI values. The RMSE was 6.11 m for the uncorrected RSSI values, 3.49 m for fingerprinting, and 2.41 m for the proposed method. The proposed method was able to effectively reduce the positioning error even in the untrained points with a maximum positioning error of 5.1 m.

Considering the use of the trilateration positioning method and covering almost 128 m^2^ lab area size, we selected four transmitters for this study. The number of transmitters by other groups has also been based on the size of their test areas. In [34], five transmitters were used for 120 m^2^ with no furniture, and in [23], four transmitters were deployed for a 100 m^2^ office area. The preference to mount the transmitters on the walls, rather than the corners, inside our test environment, was to provide a maximum propagation response in a wider direction.

In [21], with three BLE advertising channels, an error of 4.6 m, 90% of the time, was achieved in a conference room with a 16.50 m × 17.60 m size. Another study [20] with one beacon per 9 m BLE deployment achieved 2.56 m 90% of the time in the corridor area with no complex environment. In contrast, our results in a complex environment including large cabinets, chairs, desks, metal shelves, and racks (Figure 9) with a 126 m^2^ area show a 2.4 m RMSE position error when using the MLP algorithm.

## 7. Conclusions

In this paper, an artificial-based system has been proposed and implemented for detecting and correcting human body blockages in BLE RSSI values, using separate advertising channels and vision information. Sliding windows of RSSI from the three advertising channels and the number of people detected in wearable camera imagery were used as inputs for two ANN algorithms that then output corrected RSSI values. For operational system training, data should contain different time spans to improve the overall generalization. While we had limited access to the testing location, an ideal system would include routinely collected and updated training data.

The output-corrected RSSI values of both ANN methods were converted to ranges using a simple log-distance model with empirical path loss exponents found from the training data. The obtained ranges were used to compute location through trilateration. The results showed significant improvement in the range and position accuracy compared with the AI method in [24] that did not have access to vision information.

It is observed that the proposed method improves the localization solutions in complicated lab environments. The AI algorithm augmented by the vision information results provided 3.7 m position accuracy 90% of the time for the MLP algorithm, whereas the artificial-based system demonstrated 6.7 m position accuracy 90% of the time. Nevertheless, fingerprinting and classic algorithms offered 8.7 m and 12.3 m position accuracy in the same situation.

For future work, several improvements could be made to the present work. First, the proposed system should be trained and tested in more and more complicated areas and dynamic scenarios. Then, the system should be tested in additional locations without further training. Since real environments are more complicated than those tested in this paper, with real traffic and multiple obstructions, training the ANN to identify multiple human obstructions is one of the most important areas for future investigation. Identifying the BLE beacons in the imagery, and determining whether the people in the imagery are in LOS or not, should be also investigated. The application of an RGB-D camera to enable the detection of all possible static and dynamic obstacles with exact distance from the users to increase the accuracy of the system could be the subject of further research.

## Figures and Tables

**Figure 1 sensors-22-04320-f001:**
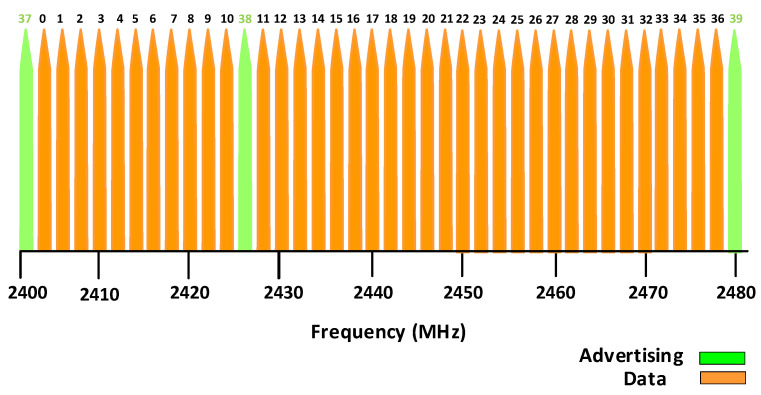
The BLE frequency channels with 37 data transmission channels (orange) and the three advertising channels (green).

**Figure 2 sensors-22-04320-f002:**
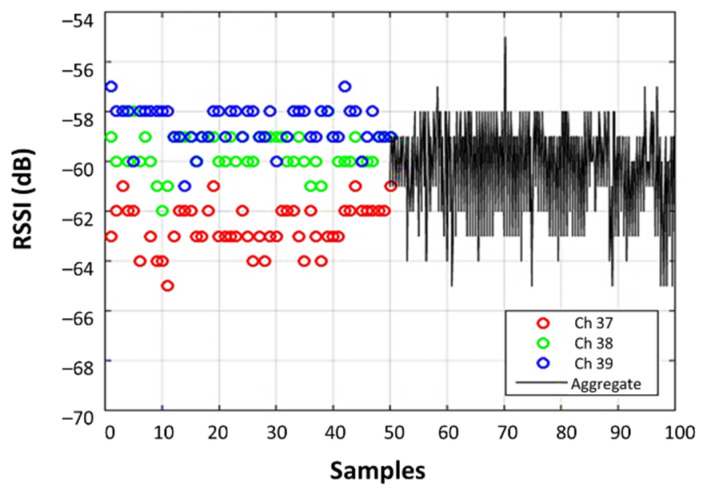
Measured RSSI samples received from the three BLE advertising channels compared to the combination (aggregate) that is often reported by BLE devices.

**Figure 3 sensors-22-04320-f003:**
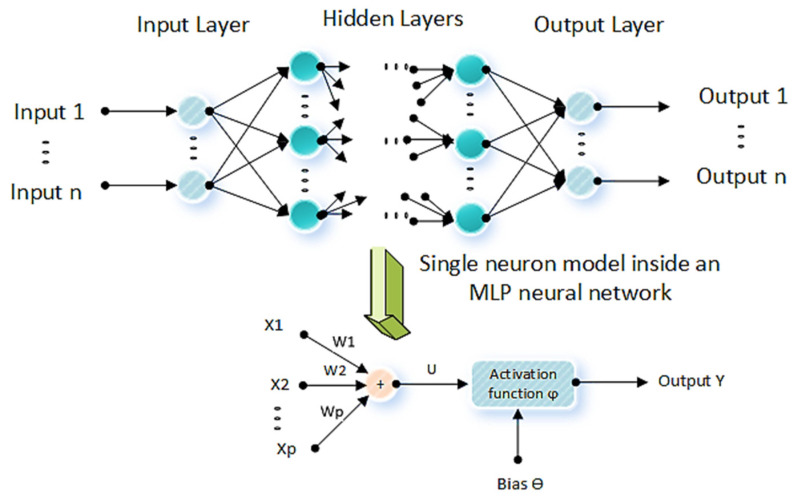
Structure of an MLP neuron network.

**Figure 4 sensors-22-04320-f004:**
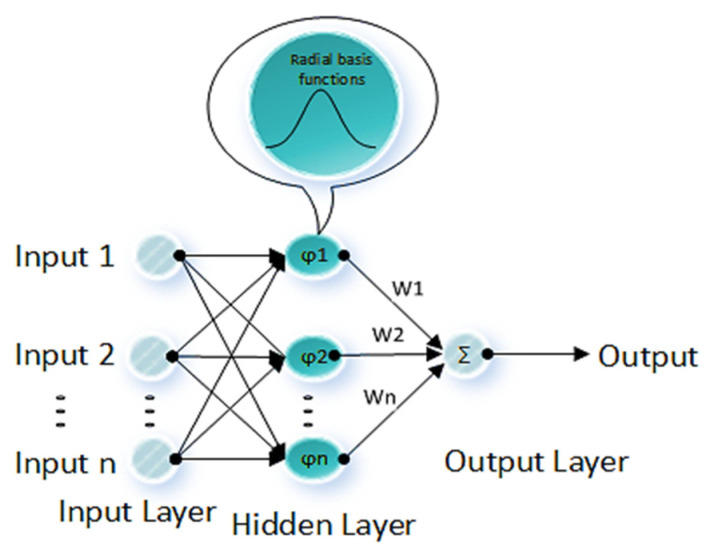
The structure of an RBF neuron network.

**Figure 5 sensors-22-04320-f005:**
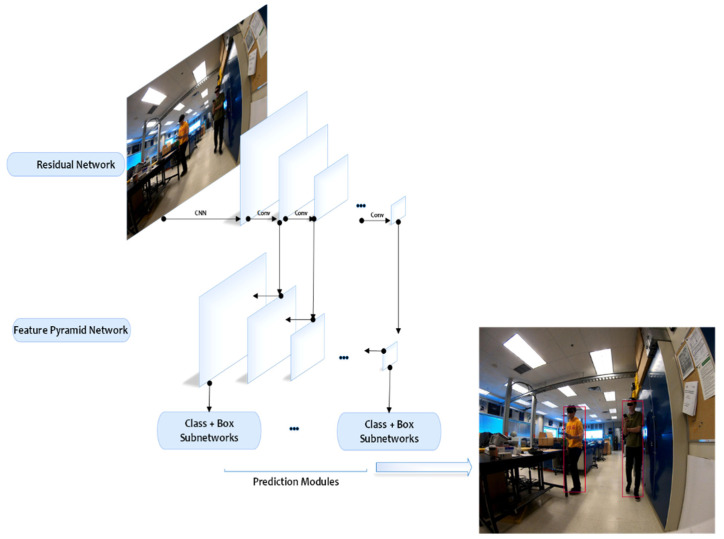
RetinaNet: the deep learning architecture adapted in the proposed framework.

**Figure 6 sensors-22-04320-f006:**
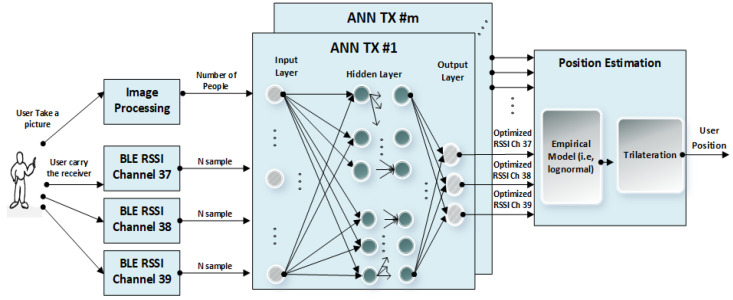
The general overview of the proposed system.

**Figure 7 sensors-22-04320-f007:**
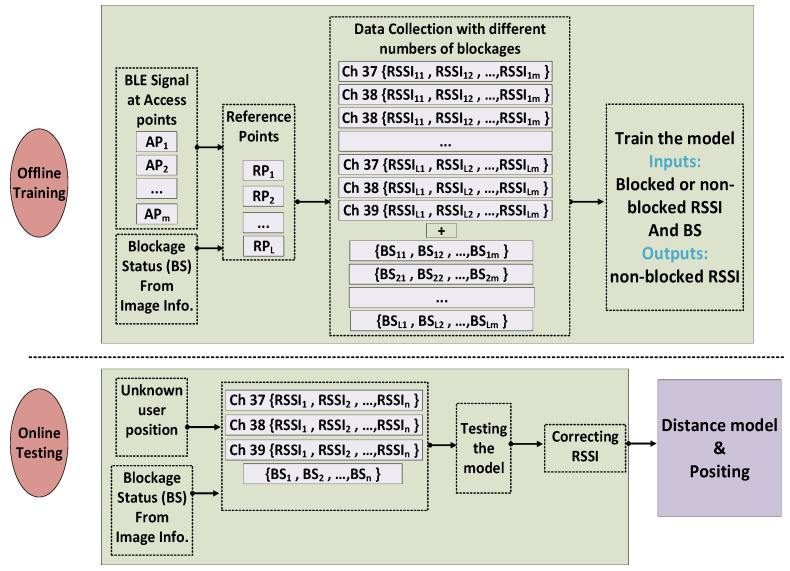
Schematic diagram of the proposed approach.

**Figure 8 sensors-22-04320-f008:**
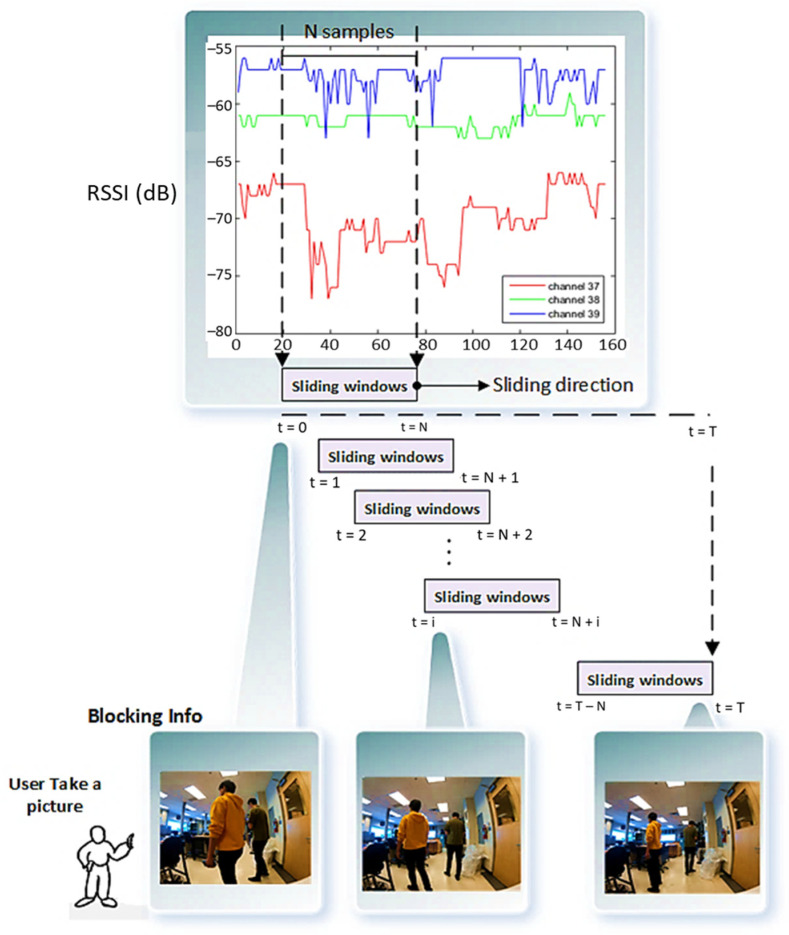
The general sampling architecture of the ANN and vision information.

**Figure 9 sensors-22-04320-f009:**
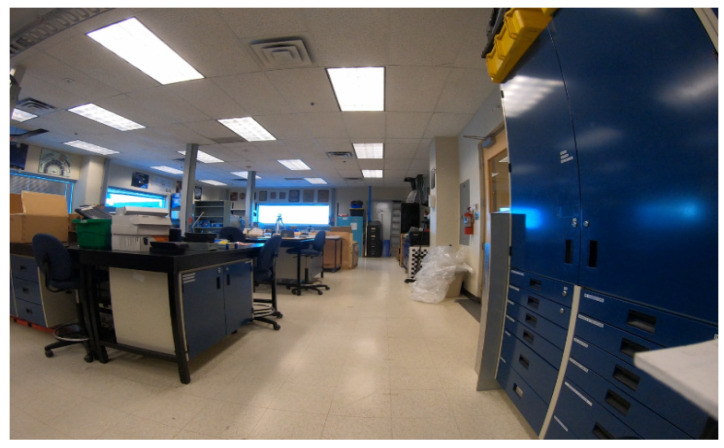
Lab area test environment.

**Figure 10 sensors-22-04320-f010:**
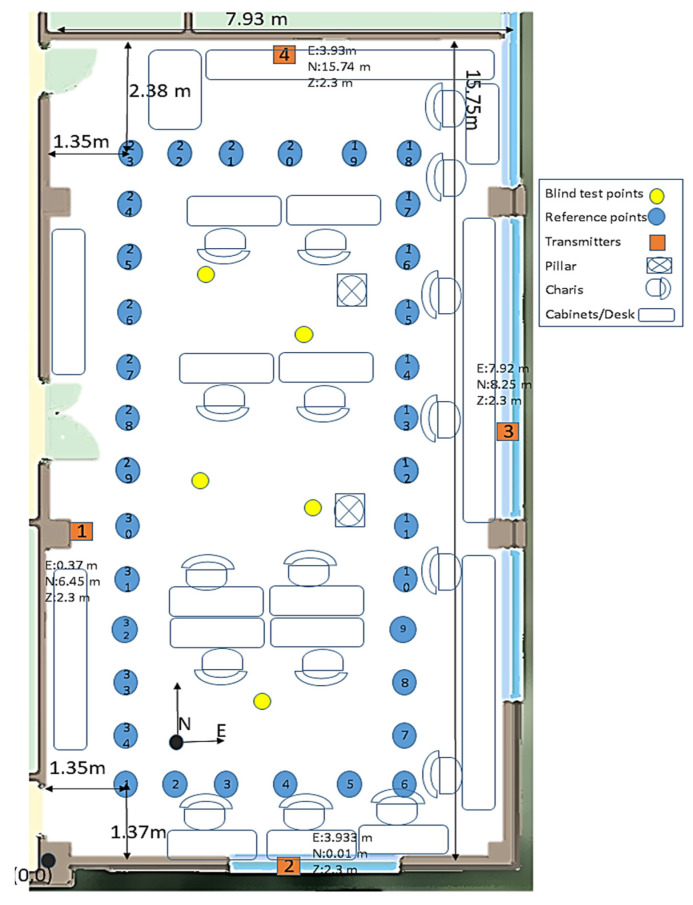
Plan view of the test environment.

**Figure 11 sensors-22-04320-f011:**
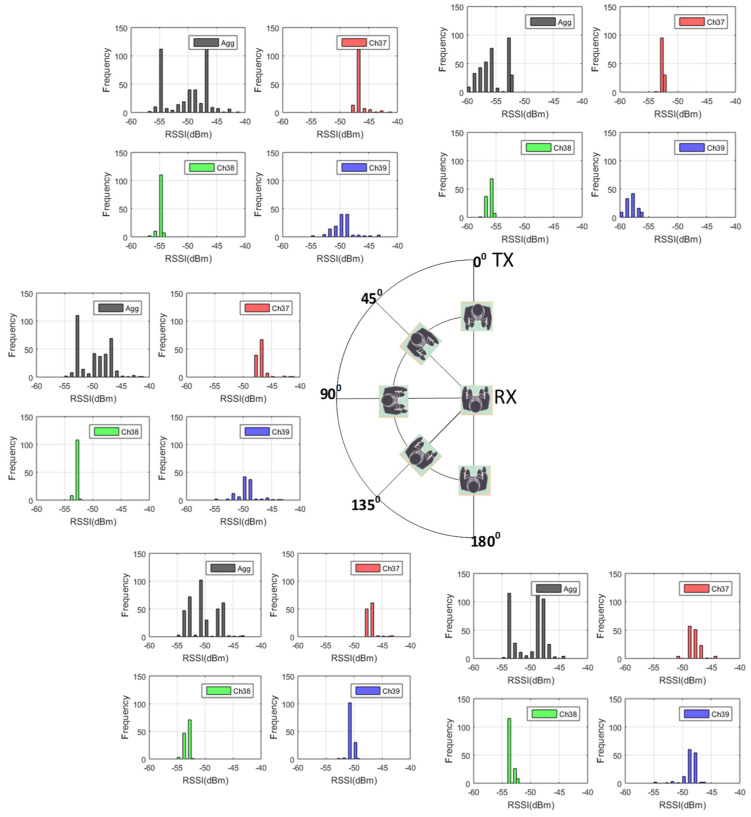
The RSSI samples received from the three advertising channels and aggregate signals in comparison with different blockage angles.

**Figure 12 sensors-22-04320-f012:**
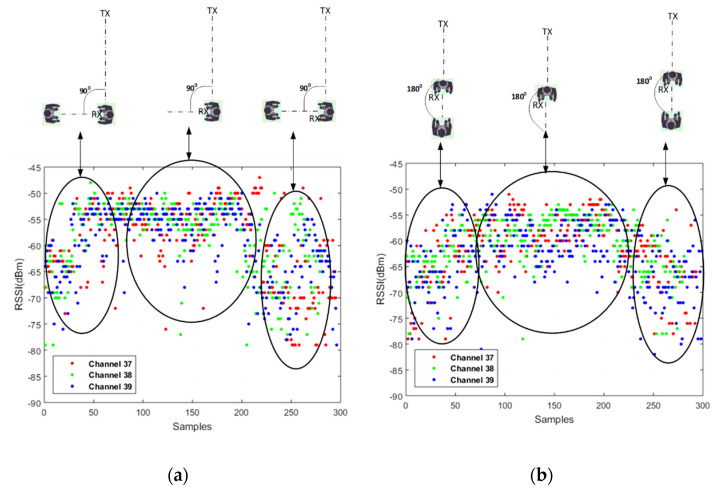
RSSI samples received from the three advertising channels in receiver facing (**a**) 90° and (**b**) 180° rotation from the transmitter.

**Figure 13 sensors-22-04320-f013:**
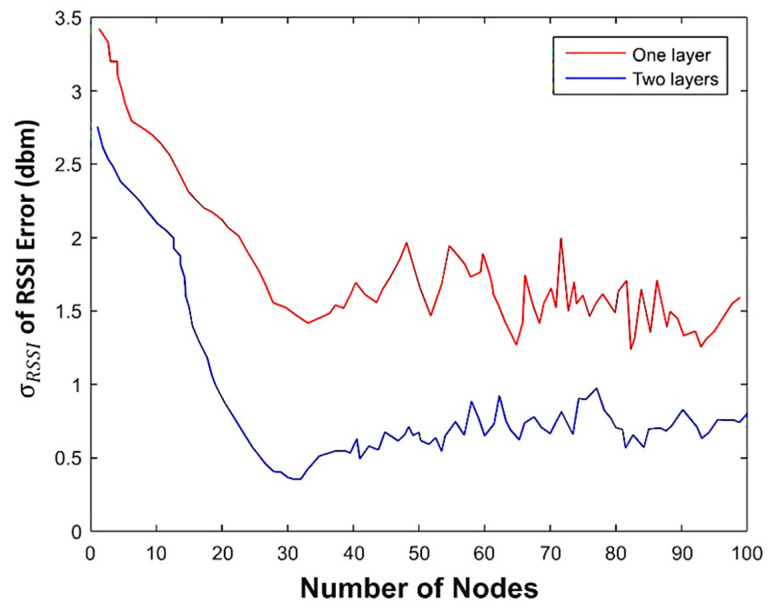
The standard deviation of the output errors of MLP in various layers and neurons.

**Figure 14 sensors-22-04320-f014:**
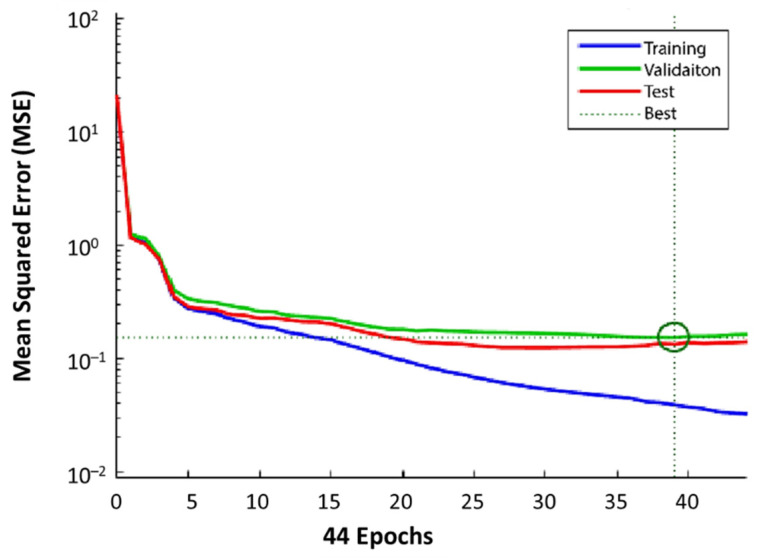
Mean squared error as a function of the number of iterations for training, validation, and testing performance in the transmitter number 1. The best validation performance is 0.15207 at epoch number 39.

**Figure 15 sensors-22-04320-f015:**
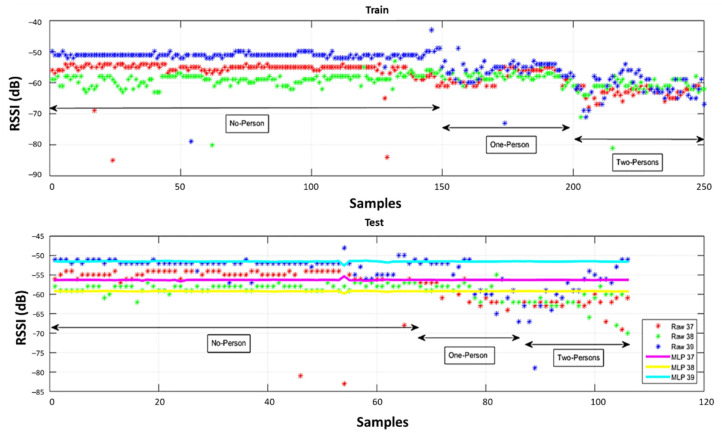
Comparison of the collected RSSI values before and after the ANNs testing and training.

**Figure 16 sensors-22-04320-f016:**
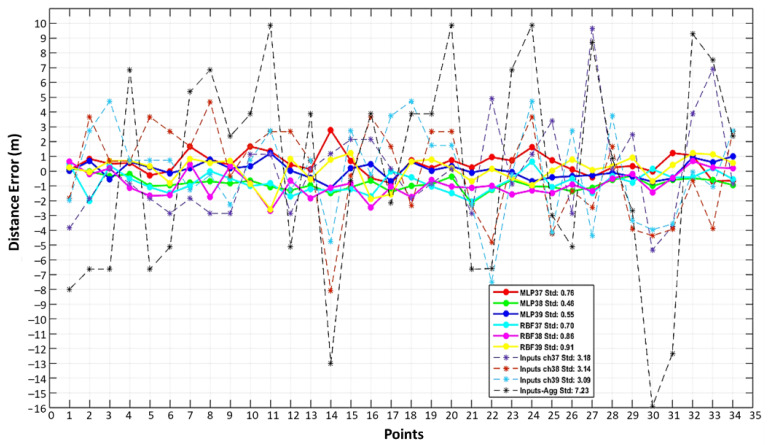
Mean of the distance error for each reference point for transmitter #1.

**Figure 17 sensors-22-04320-f017:**
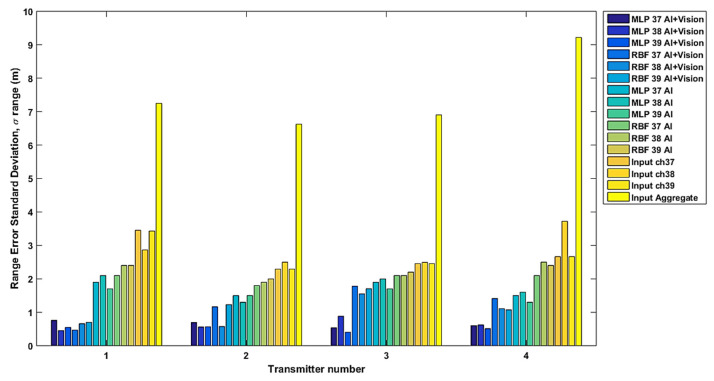
The standard deviation of the distance error before (input) and after the RSSI correction, using MLP and RBF with RSSI only (AI) [24] and using MLP and RBF with RSSI and vision information (AI+ Vision).

**Figure 18 sensors-22-04320-f018:**
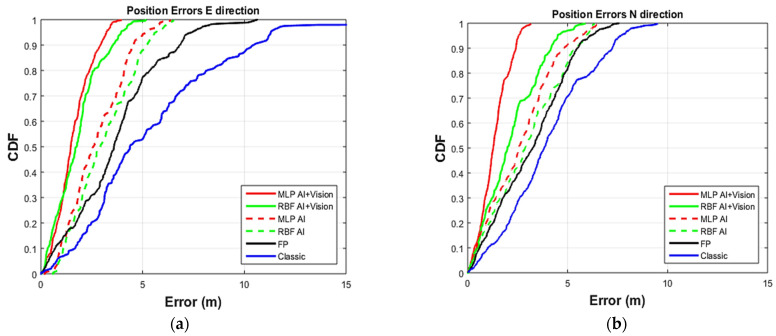
CDF of the positioning error for (**a**) east and (**b**) north directions.

**Figure 19 sensors-22-04320-f019:**
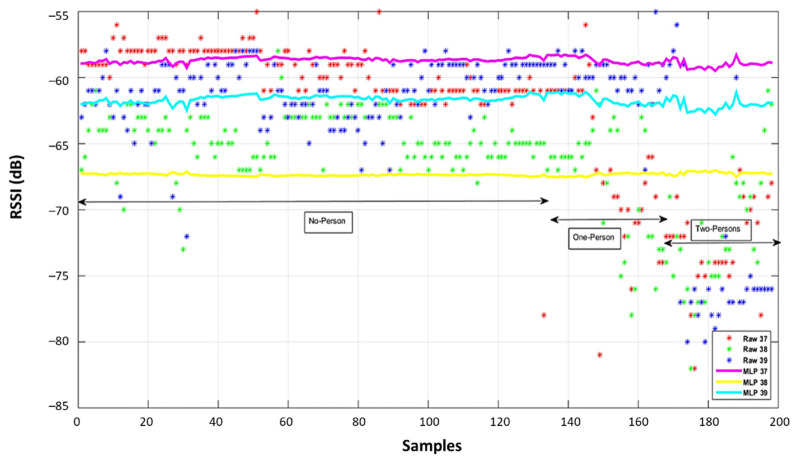
Comparison of the collected RSSI values in bind points before and after ANNs.

**Figure 20 sensors-22-04320-f020:**
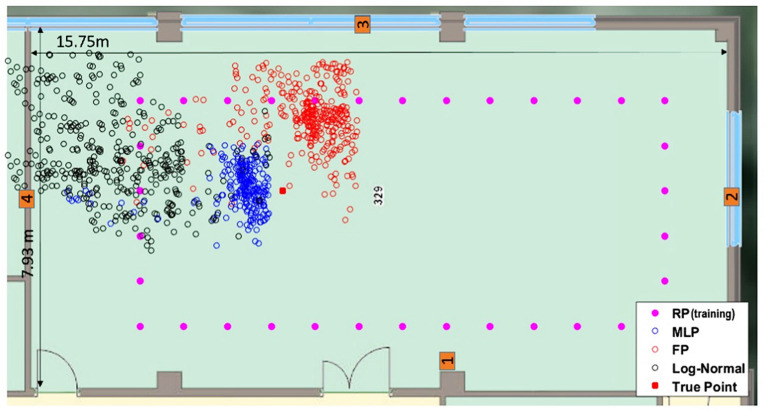
A comparison of the positioning results between the fingerprinting (FP) method, the proposed MLP algorithm with vision information, and the uncorrected log-normal method.

**Figure 21 sensors-22-04320-f021:**
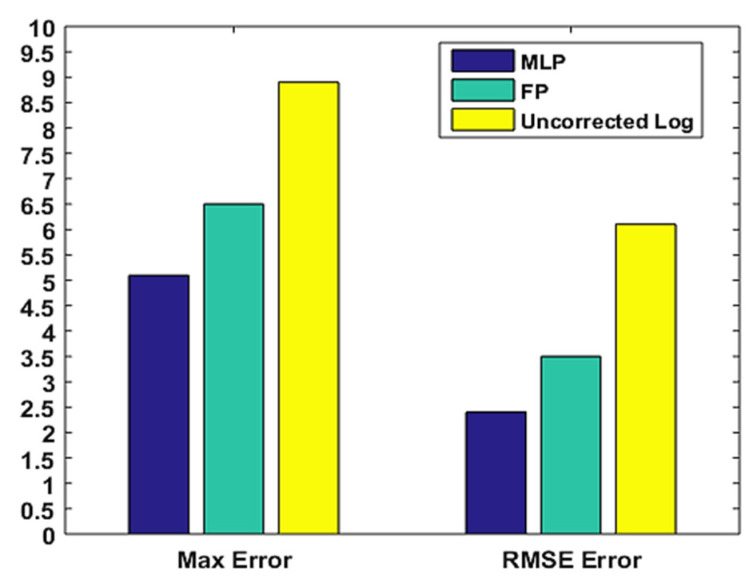
Comparison of the positioning maximum and RMSE error.

**Table 1 sensors-22-04320-t001:** Mean and standard deviation of the RSSI measurements.

Angle from LOS (Degrees)	Channel	Mean RSSI (dBm)	Standard Deviation (dBm)
0	Aggregate	−55.9	2.7
	Channel 37	−52.7	0.4
	Channel 38	−56.2	0.6
	Channel 39	−58.7	1.7
45	Aggregate	−55	3.6
	Channel 37	−46.8	1
	Channel 38	−55	0.4
	Channel 39	−49.8	1.9
90	Aggregate	−49.9	2.7
	Channel 37	−47.1	1.1
	Channel 38	−53	0.3
	Channel 39	−49.7	1.8
135	Aggregate	−50.5	2.5
	Channel 37	−47.2	0.86
	Channel 38	−53.4	0.5
	Channel 39	−50.8	0.5
180	Aggregate	−50.3	2.7
	Channel 37	−48.2	1.1
	Channel 38	−53.7	0.5
	Channel 39	−48.8	1.2

**Table 2 sensors-22-04320-t002:** Mean and standard deviation values of RSSI with and without the shadowing effect.

Angle from LOS (Degrees)	Channel	Mean (dBm)	Mean (dBm)	STD (dBm)	STD (dBm)
		With People	With No People	With People	With No People
90	Aggregate	−62.7	−56.2	5.3	5.6
	Channel 37	−68.2	−62.5	3.5	3.2
	Channel 38	−61.6	−53.1	3.9	1.8
	Channel 39	−58.1	−52.9	1.9	4.5
180	Aggregate	−69.1	−61.9	6.7	6
	Channel 37	−76.6	−67.8	4.6	4.7
	Channel 38	−66	−59.9	3.5	3.4
	Channel 39	−64.5	−58.1	3.5	4.2

**Table 3 sensors-22-04320-t003:** Vision performance in terms of the correct detection, missed detection, and false alarm.

Status	Samples	Percent
Correct detection (obstruction)	188	92%
Missed detection	16	8%
False alarm	34	10%
No detection (no obstruction)	306	90%

## Data Availability

Not applicable.

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
