# Peer review of "Combining Multichannel RSSI and Vision with Artificial Neural Networks to Improve BLE Trilateration"

_sensors, 2022, doi:10.3390/s22124320_

Round 1
Reviewer 1 Report
The article presents indoor localization approach using Bluetooth Low Energy beacons where position of the device is determined based on the received signal strength indicator (RSSI) measurements. The measurements are collected for advertisement messages in 3 radio channels used by BLE to broadcast beacons. Because RSSI measurements drop with distance form the transmiter the measurements can be used to estimate the distance and find location of the reciver. The measurements, however, are affected by propagation phenomena which cause variations in RSSI and lead to errors in distance and position estimation. Because some of phenomena result from obstacles (people) in the localization area Authors propose to use image recognition to detect people and correct the measurements accordingly. This is interesting approach, however, several technical details are not clearly presented and practical applications are vague (mainly due to energy and computational cost of image capture and processing).
Authors should be more precise in presenting the contribution of the paper. Currently it is not clear if image processing is used to detect people in the are, detect and count, or maybe localize people in relation to transmitters and the receiver? Examples presented take pictures in single direction making it impossible to detect people located in other directions from the receiver. Simultaneously, the transmitters (beacons) are located in different directions from the receiver (to improve location estimation) and obstacles towards some of them are not detect/unknown. Consequently receiver can only correct some RSSI measurements (from one beacon) while other measurements may still be affected by obstacle and it adversely affect the localization (improving measurements form one beacon does not solve the problem). It is not clear how Authors address this aspect.
It is recommended that section 1 contains clear presentation of the contribution of the article.
Related work is focused on using multiple radio channels for RSSI measurements, but it is only limited to 3 advertisement channels. Currently BLE supports advertisements in all 40 BLE radio channels which was reported to improve RSSI based localization recently [1]. It would be recommended that Authors address the aspect of using more than 3 primary advertising channels and, if possible, compare benefits of using the proposed approach vs. multiple radio channels.
It would be also beneficial if authors compare current result to the previous results presented in earlier work [2], to better present benefits of the current approach.
It is not clear how the metrics for comparison of different approaches were chosen. for example the Fig 17 presents standard deviation of the distance errors. Fig 18 compares separate North-South and East-West direction - there is no clear comparison of Euclidean positioning error for different approaches. Fig. 21 compares maximum error. The presentation of the result is inconsistent and counter intuitive.
Technical issues:
- Figures need to be in higher quality (resolution) and fit the width of the column (e.g. legend in Fig 17 is almost entirely invisible, Fig 15 is illegible)
- Axis on the figures should have units (e.g. Y axis in Fig. 14) and their description should be verified (e.g. Fig 14)
[1] M. Nikodem and P. Szeliński, "Channel Diversity for Indoor Localization Using Bluetooth Low Energy and Extended Advertisements," in IEEE Access, vol. 9, pp. 169261-169269, 2021, doi: 10.1109/ACCESS.2021.3137849.
[2] S. Naghdi and K. O’Keefe, “Detecting and Correcting for Human Obstacles in BLE Trilateration Using Artificial 671
Intelligence,” Sensors, vol. 20, no. 5, p. 1350, Jan. 2020, doi: 10.3390/s20051350.
Reviewer 2 Report
Quite nice and informative paper which: “investigate the application of simple BLE trilateration positioning in real indoor environments in which human bodies are present and can affect the signals using an ANN to correct the observed RSSI measurements”.
This paper is on trilateration positioning, not localization. Please check all references to localization (for example, line 36: RF location system rely …; line 53: … wireless ranging localization methods, line 302: The person carrying the mobile receiver to be located …), and use the appropriate terminology: positioning/positioned. Check also keyword: localization -> positioning. Check paper: https://www.isprs-ann-photogramm-remote-sens-spatial-inf-sci.net/III-4/89/2016/ (which is not written by the reviewer ..)
Line 28: Although this paper in not about GNSS, it would help to explain into some more detail why GNSS signals are not usable in a satisfactory manner in indoor environments.
Line 64: the same counts for Wi-Fi (by the way: also good to use the term: Wi-Fi, instead of WiFi).
Line 193, 196: “Trilateration algorithms use the distance ??, estimated from the received RSSI values from all transmitter nodes to compute the position of the single intersection.”. “If ? transmitters with known coordinates”. Would be nice to state one of the advantages of fingerprinting which does not require to know the position of the transmitters.
All figures: seems to be jpg-ed compressed, and very blurry. Also: figures 3 and 4 lacks a reference (not own figures).
Line 395: So, only four transmitters. Seems to be a realistic number given the size of the lab area (8 m by 16m). But … would like to see a discussion about the number of transmitters, their placement (why not in the corners of the room), the size of the lab area, and the obtained results for the proposed (RMSE Error: 2.41 m). Also: “The AI algorithm augmented by the vision information results provided 3.7 m position accuracy in 90% of the time for the MLP algorithm whereas the artificial-based system demonstrated 6.7 m position accuracy in 90% of the time. Nevertheless, fingerprinting and classic algorithms offered 8.7 m and 12.3 m position accuracy in the same situation.”
RMSE of 2.42, 6.7, 8.7 or even 12.3 are not very useful results for (exact) positioning. Might be even better to go for a kind of localisation: the proposed method will provide a proper solution to indicate the receiver is in which part of the room (for example the North-West area)
Reviewer 3 Report
My comments are attached.

Round 2
Reviewer 3 Report
The authors have addressed all of my comments. I recommend this paper in its present form.